# Coping with Trauma and Symptoms of Post-Traumatic Stress Disorder: Exploring Intentions and Lay Beliefs about Appropriate Strategies among Asylum-Seeking Migrants from Sub-Saharan Africa in Germany

**DOI:** 10.3390/ijerph19031783

**Published:** 2022-02-04

**Authors:** Freyja Grupp, Marie Rose Moro, Sara Skandrani, Ricarda Mewes

**Affiliations:** 1Division of Clinical Psychology and Psychotherapy, Department of Psychology, University of Marburg, 35032 Marburg, Germany; gruppf@uni-marburg.de; 2AP-HP, Hospital Cochin, University of Paris, Unité Inserm 1018, CESP, 75014 Paris, France; marie-rose.moro@aphp.fr; 3Hospital Cochin Paris, University of Paris Nanterre UR4430, 75014 Paris, France; saramarie.skandrani@parisnanterre.fr; 4Outpatient Unit for Research, Teaching and Practice, Faculty of Psychology, University of Vienna, 1010 Vienna, Austria

**Keywords:** asylum-seekers, coping, mixed-methods, post-traumatic stress disorder, refugees, trauma

## Abstract

Asylum-seekers are at high risk of developing post-traumatic stress disorder (PTSD) due to frequent exposure to trauma. We investigated the coping intentions and lay beliefs about appropriate coping strategies among asylum-seekers from Sub-Saharan Africa in Germany. The study applied a methodological triangulation strategy with a vignette describing symptoms of PTSD. In a quantitative part, asylum-seekers (n = 119) that were predominantly from Eritrea (n = 41), Somalia (n = 36), and Cameroon (n = 25), and a native comparison sample (n = 120) responded to questionnaires assessing coping, traumatic events, and post-traumatic symptoms. In a qualitative part, asylum-seekers (n = 26) discussed coping strategies in focus groups. In the quantitative part, asylum-seekers displayed higher intentions for religious coping, emotional support, and denial compared to the native participants. Asylum-seekers with a higher symptom load expressed lower intentions to seek instrumental support. Asylum-seekers with a lower educational level and those with a higher symptom load expressed higher intentions for substance use. In the qualitative part, we identified three superordinate themes: (a) religion, (b) social support systems, and (c) cognitive strategies. Asylum-seekers expressed coping intentions that are associated with an adaptive response to trauma. Less-educated asylum-seekers with a higher symptom load might constitute a particularly vulnerable group.

## 1. Background

The deleterious psychological effects of exposure to trauma and hardship before, during, and after the flight of asylum-seeking migrants from Sub-Saharan Africa often leads to a high prevalence of post-traumatic stress disorder (PTSD). Past research in asylum-seeking and refugee populations from Africa found PTSD rates of up to 79% [1]. Moreover, this population appears to be at a particularly high risk for ongoing post-migration stress [2]. According to the World Health Organization [3], PTSD is defined by the core symptoms of intrusiveness or re-experiencing the trauma (nightmares, flashbacks, and recurring memories); hyperarousal (difficulty sleeping, irritability, hypervigilance); and avoidance (reminders of events, dissociation) [4]. Studies have shown that while exposure to trauma is a major risk factor for PTSD, some individuals do not go on to develop PTSD following traumatic experiences [5,6]. Scholars have attributed this, inter alia, to different strategies that are used to cope with potential traumatic events and stressors [6,7]. Coping can be defined as “thoughts and behaviors that are used to manage the internal and external demands of situations that are appraised as stressful”, and distinct coping strategies have been described within the context of traumatic stress [8]. Therefore, researchers have begun to explore the coping strategies of refugees and asylum-seekers after the experience of trauma and hardship, and have found that individual, cultural, or contextual differences in coping strategies may contribute to whether or not individuals develop symptoms of PTSD [9]. Through the lens of an individualistic Western psychological perspective, scholars have deemed active or instrumental coping strategies, such as positive thinking or dealing actively with problems, to be adaptive strategies, while passive coping strategies, such as avoidance or disengagement, are mainly considered as maladaptive [6,7]. 

However, past research on coping strategies among refugees and asylum-seekers from different Sub-Saharan African origins has reported that the most common coping strategies include faith and religion, social support, and various cognitive strategies [10,11]. An abundant body of research on African refugees and asylum-seeking migrants from different countries of origin has emphasized the role of spirituality and the commitment to a religion as an essential coping strategy in overcoming trauma and hardship [12,13,14]. In this regard, scholars have described belief systems in general, including religious, social, or political beliefs, as powerful shared strategies that allow individuals to better process traumatic experiences and to better cope with the symptoms of PTSD [15,16]. Moreover, extensive research has found the reliance on social networks such as family, friends, and the community to be an important coping strategy in refugees and asylum-seekers from Sub-Saharan Africa [17,18]. With regard to cognitive coping strategies, studies with unaccompanied minors from Sudan pointed out the utility of cognitive coping strategies such as reframing the situation and focusing on the future, wishes, hopes, and aspirations [16]. In another study, the participants reported the development of an inner strength and resourcefulness, as well as normalization and minimization of the severity of the situation, as coping strategies [17]. Further reported coping strategies include the suppression of traumatic memories and using distractions such as sleeping, reading, playing a game, and talking to friends to avoid difficult thoughts and feelings. [17,19]. 

In general, scholars have assumed that cultural and contextual differences between asylum-seeking migrants and native populations regarding the ways of coping with adversity, but empirical evidence is generally lacking [20]. To improve the prevention and treatment of PTSD among asylum-seeking migrants from Sub-Saharan Africa, we argue that further research is needed to gain an understanding of coping intentions and lay beliefs regarding appropriate coping strategies. To the best of our knowledge, most of the research that has been conducted to date has been qualitative in nature, and the few quantitative studies did not include native comparison groups. Therefore, it is unknown whether there are quantitative differences in coping intentions between asylum-seeking migrants from Sub-Saharan Africa and Western host populations. Furthermore, we are not aware of any studies investigating the link between the symptoms of PTSD and coping intentions among asylum-seekers from Sub-Saharan Africa. Therefore, the purpose of this study was to explore beliefs about appropriate strategies to cope with trauma and symptoms of PTSD among asylum-seekers of predominantly Eritrean, Somalian, and Cameroonian origin in Germany, and their relationship with trauma exposure and symptoms of PTSD. To address this, we used a qualitative and quantitative methodological triangulation strategy. The triangulation of the data that was drawn from different sources allows researchers to intimately explore constructs within cultures and generalize these constructs to the same cultural context. We argue that the findings of each of the employed methods complemented each other and provided a more complete and contextual sensitive understanding of the investigated phenomena [21]. 

In the first quantitative part of the study, we compared the coping intentions of asylum-seeking migrants of predominantly Eritrean, Somalian, and Cameroonian origin that were residing in Germany with those of German participants without a migration background. In the second qualitative part, we conducted focus group discussions with asylum-seekers to explore their beliefs about appropriate coping strategies in a more profound and contextually sensitive manner. 

## 2. Materials and Methods

### 2.1. Procedure

The study is part of a larger research project examining the different aspects of explanatory models for PTSD symptoms in asylum-seeking migrants from Sub-Saharan Africa. The study consisted of two parts: a quantitative questionnaire survey and a qualitative focus group study (for a detailed description of the procedure see also [22,23]. Data were collected in Germany between April and November 2016. Ethical approval was obtained from the local review board of the Department of Psychology, University of Marburg, Germany, and written informed consent was obtained for each part of the study.

The present study is placed within an illness explanatory model framework using a vignette methodology, promoting the perspectives of laypeople [24]. In this regard, we aimed to explore culturally and socially shared perspectives regarding strategies that are perceived as appropriate in coping with the symptoms of PTSD [see also [24]]. However, with regard to the participants that were seeking for asylum in Germany, given the high prevalence of PTSD, we assumed that a large proportion of the participating asylum-seekers had either experienced symptoms themselves or knew friends or family with experiences of PTSD. 

We decided to target asylum-seeking migrants from Sub-Saharan African countries, as the numbers of African migrants seeking for asylum in Germany have remained persistently high since 2009, with Eritrean and Somalian citizens being among the ten nationalities with the highest number of entries [25].

In the quantitative part, a sample of asylum-seeking migrants (n = 119) from seven Sub-Saharan African countries, mainly Eritrea (n = 41), Somalia (n = 36), and Cameroon (n = 25), and a comparison sample of Germans without a migration background (n = 120) took part in the survey. The survey was available in German, English, French, Tigrinya, and Arabic, and was disseminated in a paper-and-pencil form and an online version. Contacting the participants required ethnographic fieldwork by the first author in places where asylum-seekers are commonly located in accommodation facilities [26], by networking at cultural gatherings, or through collaboration with civic refugee initiatives. In the quantitative part of the study, a combination of convenience and snowball sampling was employed. While we are aware of the reduced representativeness of the sample that is caused by the snowball method, our own previous research experiences, as well as that of other researchers [27], showed the utility of the method for this particular population. In the quantitative part of the study, the participants had the opportunity to express their interest in taking part in a subsequent interview study. As is common in qualitative research, purposive sampling was utilized to contact the interested participants for the subsequent qualitative part of the study. As the quantitative sample consisted mainly of individuals from Eritrea, Somalia, and Cameroon, the focus groups were sampled according to these countries of origin. In the qualitative part, asylum-seekers (n = 26) discussed their perspectives regarding appropriate coping strategies for trauma and PTSD in focus group discussions. In total, eight focus groups were conducted, with participants from Eritrea (three focus groups; n = 10), Somalia (three focus groups; n = 8), and Cameroon (two focus groups; n = 8). The lead author FG moderated the focus group discussions with participants from Somalia and Cameroon in English and French. The focus group discussions with participants from Eritrea were carried out with instructed bilingual Tigrinya to German interpreters. 

The inclusion criteria for study participation were as follows: All participants had to be aged 18 years or over, the asylum-seekers were required to have an origin in a Sub-Saharan African country and a refugee background, and the participants in the group without a migration background had to have been born in Germany and to have no migration background. 

### 2.2. Measures and Materials

In each part of the study, we asked the participants for demographic information and gave them a standardized vignette (see Appendix A) illustrating a hypothetical friend describing symptoms of PTSD according to criteria that are outlined in the International Classification of Diseases, Tenth Revision (ICD-10; [3,28] Neither the term trauma nor PTSD was used. As a triggering event for the described symptoms, a stressful event of exceptionally threatening nature was indicated, according to criterion A of the ICD-10 classification [3]. We adjusted this event for asylum-seekers and participants without a migration background to increase the possibility that they had experienced such a traumatic event; we considered “flight” to be a representative traumatic event that was experienced by asylum-seekers, we chose physical violence, operationalized as “robbery”, as the event in the vignette for participants without a migration background. We asked the participants to imagine the scenario and to indicate their coping intentions and beliefs about appropriate coping concerning the described situation. 

We translated the measurement instruments (see below) using the forward-backward translation method that was described by Flaherty et al. [29].

#### 2.2.1. Brief COPE

We assessed the coping intentions using the 28-item, English-language Brief COPE [30], which measures coping strategies on 14 scales (self-distraction, active coping, denial, substance use, use of emotional support, use of instrumental support, behavioral disengagement, venting, positive reframing, planning, humor, acceptance, religion, and self-blame). Some of these strategies are considered to be generally adaptive and others to be rather maladaptive [30]. In the present study, the standard instruction was modified to “What would you do in your friend’s place if you had these complaints?”, and the participants were asked to rate their agreement with each item on a 4-point Likert scale ranging from: 1 (“not at all”) to 4 (“a lot”). The instrument had to be translated into Arabic and Tigrinya; a French [31] and a German version [32] were already available. Psychometric characteristics were assessed for each subscale of the Brief COPE. With regard to the sample of Germans without a migration background, the internal consistency (Cronbach’s alpha) for all the scales was acceptable, with the exception of one scale (α_Planning_ = 0.38). While internal consistency for some of the scales in the sample of asylum-seekers was still acceptable, it was no longer acceptable for other scales (α_Self-distraction_ = 0.33, α_Behavioral disengagement_ = 0.34, α_Venting_ = 0.24, α_Positive reframing_ = 0.28, α_Self-blame_ = 0.28). As these scales were considered to be psychometrically unsound, they were excluded from further analyses.

#### 2.2.2. Post-Traumatic Stress Diagnostic Scale (PDS)

To assess the number of experienced traumatic events and the severity of post-traumatic symptoms, we used a modified version of the Post-traumatic Stress Diagnostic Scale PDS [33]. The list of potentially traumatic events (23 items) was extended by items from the Harvard trauma questionnaire [34] which are frequently experienced by asylum-seekers. To determine the symptom severity, the participants rated 17 items representing the cardinal symptoms of PTSD that were experienced in the past 30 days on a 4-point scale. These ratings summed up to a score ranging from 0 to 51, with the following cut-offs: 1–10 mild, 11–20 moderate, 21–35 moderate to severe, and >36 severe symptoms [35].

#### 2.2.3. Short Explanatory Model Interview (SEMI)

The focus group discussions were moderated with the help of key questions from the SEMI [36], a short interview using open-ended questions to elicit explanatory models. In the present study, we particularly concentrated on the intentions and beliefs about appropriate coping strategies. The moderator encouraged the participants to talk openly about their attitudes and experiences with the aim of eliciting concepts held, and their relationship to the current situation, context, and culture. Probes were employed to confirm the concepts that were mentioned and to explore areas of interest not raised spontaneously [36].

## 3. Analyses

### 3.1. Statistical Analysis

Statistical analyses were conducted using IBM SPSS Statistics version 25.0, with the significance level set at *p* < 0.05 (two-tailed). We investigated group differences in sociodemographic variables, traumatic events, and post-traumatic symptom load using Chi-square tests and *t*-tests. Initial correlation analyses and *t*-tests were conducted to determine the significant associations between different sociodemographic variables, traumatic events, and PTSD symptoms on the one hand, and the coping intentions on the other hand. 

The sample differences on the Brief COPE scales were analyzed using a one-way between-groups multivariate analysis of covariance (MANCOVA). We used the nine remaining scales of the Brief COPE questionnaire as dependent variables, and sample (asylum-seekers or Germans without a migration background) as the independent variable. Since significant sample differences emerged with regard to gender and age, these covariates were included in the analytical model. Furthermore, initial correlation analyses and *t*-tests revealed significant associations between religion, education (number of years and attainment), traumatic experiences, and symptom load on the one hand, and coping intentions on the other hand. Therefore, we included these variables as covariates in the analytical model. Moreover, we carried out analyses to explore the differences between the three main countries of origin of asylum-seekers: Eritrea, Somalia, and Cameroon. 

In a prior assumption testing, we checked the data for normality, homogeneity of variance-covariance matrices, and multicollinearity. We generated a correlation matrix to verify the absence of multicollinearity. Shapiro–Wilk tests revealed significant deviations from the normal distribution in some items of the Brief COPE questionnaire. However, we decided to conduct the analyses of variance, as past research [37] suggests that an ANOVA is robust to deviations from the assumption of normally-distributed dependent variables. We display the effect sizes as partial eta-squared as proposed by Cohen (1992), where 0.01–0.06 = small effect, 0.06–0.14 = moderate effect, and a value over 0.14 = large effect [38].

### 3.2. Qualitative-Analysis

The focus group discussions were audio-recorded and transcribed verbatim before being subjected to Interpretative Phenomenological Analysis (IPA; [39]. IPA was specifically developed to allow an exploration of idiographic subjective experiences and, more specifically, of social cognitions [40]. IPA acknowledges that the researcher’s engagement with the participant’s text has an interpretative element and assumes an epistemological stance, whereby through careful and explicit interpretative methodology, it becomes possible to access an individual’s cognitive inner world [40]. As such, it is concerned with the detailed examination of personal lived experience, the meaning of experience to individuals, and how individuals make sense of that experience [41]. 

The first author, who is trilingual in German, English, and French, undertook the transcriptions according to the guidelines that were proposed by Kuckartz [42], which specify how spoken language should be transcribed into written form. With regard to the degree of precision, we opted for a relatively simple transcription system, as interpretative analyses focus rather on the meanings of statements than the specific use of language. To organize and manage the data, she used the analysis software MAXQDA© version 12. All the transcripts were individually analyzed and coded using MAXQDA©. Since more data was collected during the focus groups than was relevant for answering the present research question, the transcripts were first sorted thematically by section and only the data that were relevant for answering the research question were included in the analysis. The analysis of the sections was then conducted according to the procedure that was outlined by Smith [43]. Throughout the analysis, we aimed to identify at the same time, the statements of the individual person and the narrative of the entire group to find a balance between personal and shared meaning-making. A detailed analytic treatment of each case was followed by the search for patterns across cases, resulting in the presentation of shared themes [41]. Following the guidelines that were proposed by Smith and Osborn [39], the first author began with writing comments on topics that were relevant to the research question freely and without rules by recording initial impressions. The focus was particularly on the similarities and differences of what was said, on repetitions and contradictions. Subsequently, each transcript was examined individually and read again; this time, the comments were rewritten into precise themes at one higher level of abstraction that summarized the essential points of what was said. Several themes were connected and interlinked and superordinate and subthemes were developed for each transcript, and afterwards the patterns were established across transcripts [39]. During the whole process, care was taken to ensure that the emergent superordinate themes did not lose their direct relationship to what was said. Finally, the superordinate and subthemes were thus transformed into a narrative description. All of the themes contained verbatim quotations from the transcripts and were supplemented with interpretive elements. The audited themes were reviewed with other researchers and migrants from the respective countries of origin to ensure that the conclusions were drawn in a culturally and contextually sensitive manner. 

## 4. Results of the Quantitative Part of the Study

### 4.1. Sample

A convenience sample of n = 120 German participants without a migration background and n = 119 asylum-seekers from seven Sub-Saharan African countries (Eritrea (n = 41), Somalia (n = 36), Cameroon (n = 25), Ethiopia (n = 7), Nigeria (n = 4), Togo (n = 4), and Sudan (n = 2)) participated in the study. The asylum-seekers’ mean duration of stay in Germany was approximately two years (s.d. = 2.7). A total of 71% of the asylum-seekers were male, and their mean age was 28 years (s.d. = 7.8). The participants without a migration background consisted of 35% male participants with a mean age of 37 years (s.d. = 16.5). There were significant differences between the two groups with regard to gender (χ^2^ (1, *N* = 236) = 30.13, *p* < 0.001) and age (*t* (170.36) = −5.56, *p* < 0.001). Further group differences were observed regarding religious affiliation (χ^2^ (4, *N* = 235) = 63.67, *p* < 0.001). Over half of the investigated asylum-seekers were Christians (56.3%) and one third were Muslims (33.6%), while two thirds of the Germans without a migration background were Christians (65.8%) and almost one third had no religion (31.7%). The group of asylum-seekers indicated stronger faith (*t* (151.47) = 10.65, *p* < 0.001) and perceived higher assistance from their faith compared to the Germans without a migration background (*t* (193.23) = 10.74, *p* < 0.001). Overall, the asylum-seekers had a lower educational level than did the participants without a migration background (9.7 vs. 12.2 years of formal education). On average, the asylum-seekers had experienced five different traumas (s.d. = 4.91), whereas the Germans without a migration background had experienced only one trauma (s.d. = 2.03; χ^2^ (17, *N* = 239) = 63.34, *p* < 0.001). Moreover, the asylum-seekers showed a significantly higher post-traumatic symptom load (*t* (210) = 5.88, *p* < 0.001; see Table 1). Around 74% of the asylum-seekers and 54% of the participants without a migration background had experienced at least one traumatic event. 

### 4.2. Coping Intentions—Results of the Questionnaire Survey

Compared to Germans without a migration background, the asylum-seekers stated a higher intention to use religion (large effect size) to cope with the symptoms of PTSD (Table 2). In this regard, participants from Eritrea and Somalia indicated a significantly higher intention to take recourse to religion as a coping strategy, compared to participants from Cameroon. The asylum-seekers stated a higher intention to use emotional support as a coping strategy compared to the native German participants (medium effect size). Moreover, asylum-seekers expressed higher intentions for denial than did the Germans without a migration background (large effect size). The intention to use denial as a coping strategy was stronger in asylum-seekers with a lower symptom load and in the participants without a migration background who had a higher symptom load. Muslim asylum-seekers with a higher educational level expressed higher intentions to use acceptance as a coping strategy, whereas German participants without a religious affiliation and with a higher educational level indicated lower intentions to use this strategy. Asylum-seekers who had experienced more traumatic events and had a higher symptom load expressed lower intentions to look for instrumental support to cope with the symptoms of PTSD (Table 3). Moreover, asylum-seekers with a lower educational level and with a higher symptom load expressed higher intentions to employ substance use as a coping strategy.

## 5. Results of the Qualitative Part of the Study

### 5.1. Sample

The sample of the qualitative study part comprised n = 26 that were predominantly male (89%) participants with a mean age of 27 years and a mean of 8.5 years of formal education. At the time of the study, their mean duration of residence in Germany was around 1.3 years. Participants from Eritrea (n = 10) and Cameroon (n = 8) were Christians, whereas participants from Somalia (n = 8) were Muslims. The participants from Eritrea had a mean age of 31.9 years, participants from Somalia had a mean age of 25.6 years, and the participants from Cameroon had a mean age of 23.4 years. The number of years of formal education were as follows: the participants from Eritrea 10.8 years, the participants from Somalia 5.6 years, and the participants from Cameroon 9.5 years. 

### 5.2. Beliefs about Appropriate Coping Strategies—Themes Emerging in the Focus Group Discussions

The participants immediately associated the case vignette with their personal traumatic experiences, their difficult living situations, and their personal ways to cope with these. They reported several coping strategies and stated various beliefs regarding appropriate coping strategies, which we grouped into three superordinate themes: (a) religion and faith, (b) social support systems, and (c) cognitive strategies.

#### 5.2.1. Religion and Faith

The participants (n = 23) across different religious affiliations strongly emphasized that religion and faith would be the most frequently applied and effective coping strategy with regard to the described symptoms of PTSD. Furthermore, religion and faith appeared to be the common underlying and interconnecting element with regard to the other social and psychological coping strategies that were discussed. The participants expressed that an unshakable faith in God would assist in overcoming the described symptoms, and they emphasized their own firm reliance on God’s help. The participants believed in a higher plan for them and that by believing and praying, their situation could improve. This “religious meaning-making” enabled the participants to endure hardships and to overcome traumatic experiences in their past and during their migratory trajectories. 

“*The most important thing for me is faith. (…) Faith was leading me over the Mediterranean Sea. Because I am deeply faithful, I climbed onto that timber, onto boats. The strength lies in the faith.*” (male, 35 years, Eritrea)

“*You don’t protect yourself on your own. It’s God who protects us*” (male, 20 years, Eritrea)

“*Reassurance. It lets us believe that tomorrow things will be better. So for us it [religion] is really, really important.*” (male, 23 years, Cameroon)

Uprooted from their familiar context, cultural surroundings, and social frameworks, religion assumed the role of a foundation and value framework in a completely unfamiliar society. Living in accordance with religious commandments and saying prayers were described as important aspects in structuring and organizing everyday life. Praying and reading Holy Scriptures were perceived as the most helpful way of coping (n = 23). Membership of a religious group, affiliation with religious communities, and finding support within them, were described as crucial in overcoming and dealing with trauma and symptoms of PTSD. 

#### 5.2.2. Social Support Systems

During the focus group discussions, participants (n = 26) frequently referred to themselves and members of their community as “Africans”. They often made use of their collective self and their membership within a social collective by referring to “we” instead of “I” when expressing their opinion. 

“*I told them that we are Somalis. (…) Somalis is like people (…) they call each other. They wait for each other.*” (male, 30 years, Somalia)

The participants (n = 23) across the focus group discussions emphasized social support and assistance from family, friends, and their own community as important coping strategies with regard to the described symptoms of PTSD, but also for handling difficult situations in general.

“*The suffering and the problems can be minimized in exchange with friends and the community.*” (male, 31 years, Eritrea)

“*It’s an illness and in any case he needs other people who can rest with him.*” (male, 28 years, Cameroon)

“*Contact to other persons plays an important role, of course. And when you withdraw yourself from others and isolate yourself, this is something that will lead you to problems.*” (male, 25 years, Eritrea)

Even though social support was perceived as helpful and important in overcoming suffering and difficult situations, participants from Eritrea (n = 6) especially expressed their difficulties in speaking openly about problems and about their mental distress. 

“*It’s a condition where we just keep silent, (…) and try to battle our way on our own without telling another person about it. Nobody wants to burden someone else with it. Because we know very well that everybody has this problem. (…) It’s a culture in which you don’t communicate. You don’t talk about difficulties and problems.*” (male, 35 years, Eritrea)

Some (n = 3) even described that they reacted with social withdrawal when experiencing difficult situations. 

“*So for example what I did is, I just separated myself from the other people.*” (male, 19 years, Eritrea)

#### 5.2.3. Cognitive Strategies

##### Cognitive Disengagement

The participants from Somalia (n = 4) described applying a strategy of cognitive disengagement, which they called *neutralize* or *cool down the brain* to distance themselves from unpleasant and intrusive memories.

“*I try to cool down myself, but this thing is not fading away from my brain.*” (male, 35 years, Somalia)

##### Self-Distraction and Staying Occupied

In response to post-migration stressors in general and to the symptoms of PTSD, the participants across the focus groups (n = 21) perceived self-distraction and keeping occupied in one’s everyday life to be helpful coping strategies. Going for a walk, meeting and chatting with friends, drinking tea, playing football, watching TV, and sitting outside in the area of the train station were mentioned as commonly used distraction activities.

“*It’s like every time you go out, sit in the Bahnhof [train station]. That’s where people go. (…) Sit somewhere, drink a cup of tea.*” (male, 30 years, Somalia)

##### Having Prospects and Hope for the Future

Closely linked to being occupied in one’s everyday life, the participants (n = 8) emphasized the importance of dignified reception conditions and decent accommodation for asylum-seekers. They described that prospects of and access to education, work, and employment, and being able to learn about the local culture and language, were helping them to cope better with their uncertain residence status, traumatic memories, and mental distress. 

“*I saw myself confronted with all of the problems that were cited in this example. I speak out of experience. This leads you to lose your belief in something good. I was lucky to escape all of that because I could educate myself further in the fields of nursing care.*” (male, 35 years, Eritrea)

Others (n = 13) described the conviction of a better tomorrow, a brighter future, and oneself being stronger than the difficulties they were currently experiencing to be helpful cognitive coping strategies. Moreover, they described planning and focusing on aspirations for the future as a further useful strategy. The capacity to stay focused on one’s goals and to believe in one’s own inner strength was perceived as a crucial element in overcoming the described symptoms of PTSD, traumatic experiences, and hardships in general.

“*(…) I started to believe that I have to start a new life. (…) and then I tried to resolve it.*” (male, 19 years, Eritrea)

“*There are other people who have the capacity to keep their objective. This capacity allows them to overcome these hardships. And others, they don’t have this capacity. At a certain moment, because they are going through hard times, they lose sight. (…) And they will get these symptoms.*” (male, 23 years, Cameroon)

##### Reframing

Participants from Eritrea and Cameroon (n = 13) described the use of positive reframing, such as seeing the situation in a more positive light and looking for something good in what was happening to them. 

“*It’s true that we are lucky. We have a place where we can lay our heads down. There are others who don’t have it. There are others who have nothing to eat.*” (male, 23 years, Cameroon)

Furthermore, participants emphasized the role of being thankful (also in a religious context) for what they actually have, and expressed gratitude for having reached Europe after a long history of flight and to have made it that far. 

“*When you are in Europe a certain contentment sets in. (…) gratitude that you have made it this far.*” (male, 25 years, Eritrea)

“*But we say thank you to Germany. People are giving us a roof over our heads. (…). They allow us to go to school. They give us something to eat. They give us a dwelling. All of this ensures that everybody is not like our neighbor [who is mentally ill].*” (male, 23 years, Cameroon)

##### Acceptance and Normalization

Participants across the focus groups (n = 7) identified acceptance as a strategy that allowed them to cope with hardships, traumatic experiences, and the symptoms of PTSD. They described the normality of enduring difficulties and hardships throughout their lives and that one becomes accustomed to this by adopting an attitude of acceptance to whatever the future holds. Often, this attitude was closely linked to their faith and religious convictions and meaning-making. 

“*This whole situation is not new; we are used to enduring this.*” (female, 35 years, Eritrea)

“*I have noticed that for the Europeans, the Germans (…) little things are seen as dramatic and are resolved immediately. Discussed. Analyzed. But for us it’s a state, where we accept (…)*” (male, 35 years, Eritrea)

## 6. Discussion

We investigated the lay beliefs about appropriate coping strategies for trauma and the symptoms of PTSD among asylum-seekers of predominantly Eritrean, Somalian, and Cameroonian origin using qualitative methods. Moreover, we quantitatively compared the coping intentions of asylum-seeking migrants to those of a German population without a migration background. Descriptive analyses showed that asylum-seekers displayed strong intentions for religious coping, active coping, planning, and seeking emotional and instrumental support. Compared to the participants without a migration background, they expressed higher intentions for religious coping, emotional support, and denial. In the qualitative part of the study, the asylum-seekers reported several coping strategies that they believed to be helpful, which we grouped into three superordinate themes: (a) religion and faith, (b) social support systems, and (c) cognitive strategies. To the best of our knowledge, this is the first study investigating the coping intentions in asylum-seeking migrants from Sub-Saharan Africa combining a qualitative and quantitative research methodology, including a native comparison group. The results of both study parts consistently showed that religion and faith were crucially important coping strategies among asylum-seeking migrants of Sub-Saharan African origin. Moreover, it became apparent that religion and faith were common elements of all of the described coping strategies and served as a stabilizing framework in an alien society (see also [19]. In addition, there was a difference compared to the German sample without a migration background for whom faith and religion was a less significant coping strategy. Our results are in line with previous research which classified religious beliefs as a backbone of refugees’ coping with traumatic experiences and symptoms of PTSD and their ability to make meaning out of these [10]. A large body of literature has found that turning to religious faith is a coping strategy of major importance to refugees from Sub-Saharan Africa [14,18]. Asylum-seekers in the present sample showed high levels of faith. In this regard, religious coping was shown to be a strategy that was closely related to problem-focused coping for highly religious individuals [44], which underlines the adaptive nature of religious coping intentions that were described by asylum-seeking migrants from Eritrea, Somalia, and Cameroon. With regard to the psychosocial support of asylum-seeking migrants from Sub-Saharan Africa in countries of the global North, the present results suggest the need for practitioners to explore religious and spiritual frameworks to mobilize important religious resources in their clients to promote coping and resilience. In this manner, they can demonstrate understanding and acceptance of various spiritual contexts in a religiously sensitive way. This could, for instance, imply that practitioners respectfully address their client’s spiritual and religious needs by assessing a spiritual history and engaging in appropriate consultation with clergy [45]. Moreover, clinicians may also encourage clients to engage in communal networks that are associated with their religious congregation [46]. However, while religious sensitivity might be important, clinicians need to beware of cultural stereotyping and need to acknowledge the diversity of religious and spiritual belief systems, trying to be sensitive to each individual context.

Furthermore, the results of the quantitative and qualitative part of the present study consistently demonstrate that relying on social support systems is an important coping strategy for asylum-seekers from Sub-Saharan Africa. Scholars have described this as an active and adaptive way to cope with stress, trauma, and the symptoms of PTSD [6,47]. The present findings suggest that emotional support was more important to asylum-seekers than to participants without a migration background. This might be explained by the emphasis on group identification in Sub-Saharan cultures, where members are seen as entities that are embedded in the collectivity [48]. Asylum-seekers in the present study identified themselves strongly with their ethnic and racial group and their own position within a social framework, referring to their shared sociocultural and historical context [49]. Our findings imply that the individualistic patient-centered approach that is employed by Western-trained psychotherapists may fall short of what patients might expect from an intervention. In this regard, it might be beneficial for practitioners to include the social environment of patients to positively influence their motivation and follow-up attendance. Therefore, the development of community mental health programs might be helpful to meet the mental health needs of asylum-seekers and promote coping and resilience [50]. However, asylum-seekers from Eritrea expressed difficulties and fear of stigmatization in speaking openly with their social environment about mental distress, and some reacted with social withdrawal when experiencing symptoms of PTSD. This finding is in line with past research that was conducted by Melamed et al. (2019), who reported that under several circumstances, asylum-seekers from Eritrea in Switzerland felt uncomfortable disclosing their mental health states to other asylum-seeking migrants or other compatriots in the host country [51]. In this regard, social isolation and withdrawal were pointed out as risk factors and maladaptive styles of coping that affected resilience and increased the likelihood of developing severe symptoms [52,53]. For this reason, practitioners should be particularly attentive when encouraging patients to reconnect with their own community, to respect each individual context in an empathic and non-judgmental way.

In accordance with previous research, the asylum-seeking migrants in the present study believed in the efficacy of cognitive coping strategies, such as planning and focusing on future objectives [16,17]. These were related to the access to education, work, and employment, and to the ability to learn about the host country’s culture and language. Furthermore, asylum-seekers perceived self-distraction and keeping occupied as helpful strategies to cope with the symptoms of PTSD. These findings underline that even in the insecure process of seeking for asylum, the opportunity to participate in everyday life might have an important positive influence on the mental health of the migrants [54,55]. Therefore, the present findings hold important implications for policy, as provisionally-granted access to the labor market, language, and integration programs with local communities might promote coping and resilience and, in turn, reduce symptomatology—even in the absence of psychotherapeutic intervention [55].

With regard to the coping strategies that were deemed as maladaptive based on a Western psychological perspective, asylum-seekers expressed higher intentions for denial than did Germans without a migration background. Trying to deny the reality of the situation has been associated with an increased likelihood of experiencing ongoing distress and post-traumatic stress [6,56]. Furthermore, to the best of our knowledge, our study was the first investigating the link between symptoms of PTSD and coping intentions among asylum-seekers from Sub-Saharan Africa, which enabled us to identify two particularly vulnerable groups of asylum-seekers: Those with higher symptom load and a lower educational level expressed higher intentions for substance use as a coping strategy. Furthermore, the more traumatic experiences that the asylum-seekers had suffered and the higher their symptom load was, the lower was their intention to get help and advice from others to cope with the symptoms of PTSD. These results are in line with past research reporting reduced help-seeking intentions in asylum-seekers with higher levels of PTSD symptomatology [22,57]. Such findings might be explained by the fact that individuals with higher symptomatology show more avoidance behaviors, as they might expect that seeking help will entail them being confronted with their traumatic experiences. Moreover, individuals who intend to use more avoidant strategies to cope may be at the greatest risk of increasing PTSD symptoms [58,59]. In this regard, community mental health programs should specifically identify and target these individuals and assist them in choosing and implementing more adaptive coping strategies.

## 7. Limitations

When interpreting the present results, some methodological considerations should be taken into account. 

We investigated a heterogeneous sample of asylum-seeking migrants regarding countries of origin, ethnicity, and religion. Even though we focused on three countries in particular, and accounted for differences with regard to these countries and several sociodemographic factors, we recognize that countries are rarely homogeneous societies with uniform cultures and contexts. Therefore, we underline that the conclusions regarding the impact of culture on the participants’ coping intentions remain limited. 

The present data showed several violations of the assumption of normal distribution, and the internal consistency of certain subscales was below acceptable levels. Therefore, we were not able to consider all of the subscales of the Brief COPE. We assume that this could reflect a particularly Western perspective on coping strategies and, therefore, indicate a varying understanding of the investigated constructs. In this regard, several researchers have critically noted the development of psychological measures that are based predominantly on Western perspectives, values, assumptions, and norms [60,61]. When these measures are used in populations with non-Western backgrounds, issues of conceptual non-equivalence across cultures can decrease the validity of results [60,61]. Therefore, our results underline the importance of investigating the validity and psychometric properties of measurement instruments before drawing conclusions based on their scores. 

Moreover, we quantitatively explored intragroup differences within the sample of asylum-seekers with unequal sample sizes of each group. While equal sample size is not one of the assumptions that was made in a MANCOVA, there is a potential of reduced statistical power. Therefore, caution is warranted while interpreting the results. However, readers should be aware of the fact that these analyses are exploratory in nature only and should not be considered principal analyses. 

Due to the case vignette design of the study, we only focused on coping intentions and beliefs about appropriate coping. While participants of the focus group discussions recognized the described symptoms without any difficulty, future research might focus on the coping strategies of clinical populations. Furthermore, we did not control for prior treatment experiences, and asylum-seekers and participants without a migration background differed in terms of traumatic experiences and PTSD symptoms. 

The reader should be mindful of the fact that the data collection and moderation of focus groups were conducted by a white, female member of the majority society. This might have led to a selection bias. Moreover, some participants might have been reticent to share their opinion due to a feeling of social desirability, or due to differences in gender, race, social class, and cultural background. 

The participants of the present study were predominantly male, which largely corresponds to the German asylum statistics at that time [62]. As such, it should be noted that the present results depict a rather male perspective on different styles of coping with trauma and symptoms of PTSD. Indeed, past research have suggested gender differences in coping strategies, but such differences have not been fully examined to date [63,64]. Therefore, future studies may wish to consider investigating female samples and the intersectionality of influences of gender and cultural and sociodemographic background on coping intentions and beliefs about appropriate coping with the symptoms of PTSD. 

## 8. Conclusions

Asylum-seeking migrants from Sub-Saharan Africa, particularly Eritrea, Somalia, and Cameroon, expressed several coping intentions that were associated with a helpful and adaptive response to hardship and trauma [6,7]. Therefore, our findings indicate important resources that can be mobilized in this particularly vulnerable group. On the other hand, asylum-seekers endorsed higher intentions for denial than did participants without a migration background, which has been deemed as avoidant and mostly maladaptive based on a Western psychological perspective [6]. 

To improve the prevention and treatment of PTSD among asylum-seeking migrants, specific interventions should be tailored to asylum-seekers’ resources and coping intentions. To this end, it might be advisable for professionals that are working with asylum-seekers to encourage helpful and adaptive coping, whereas maladaptive coping strategies might be identified and modified. Moreover, clinicians should identify the relevant post-migration stressors that have significant implications for how the mental health needs of asylum-seekers can be managed [17]. In this regard, practitioners can acknowledge and focus on the impact of ongoing stressors such as difficulties adjusting to the new living conditions. Individuals who do seek out psychotherapy are unlikely to benefit from treatment while they are still struggling in an economically disadvantaged environment lacking social support [16]. As such, Khawaja et al. (2008) suggest community resources, in the form of involvement in religious and social activities that need to be utilized to address ongoing post-migration stressors such as social isolation and poor language skills. 

Particular consideration, in this respect, should be dedicated to asylum-seekers with a lower educational level and asylum-seekers with a higher symptom load, as they might constitute a particularly vulnerable group.

## Figures and Tables

**Table 1 ijerph-19-01783-t001:** The distribution of the experienced and witnessed traumatic events and the Post-traumatic Stress Diagnostic Scale (PDS) levels of post-traumatic symptoms *.

	Germans without a Migration Background	Asylum-Seekers	Eritrea	Somalia	Cameroon
	*M* (*SD*)	*M* (*SD*)	*M* (*SD*)	*M* (*SD*)	*M* (*SD*)
Number of traumatic events experienced or witnessed	3.5 (3.05)	7.8 (6.17)	8.6 (6.3)	8.4 (6.2)	6.2 (6.4)
PDS symptom severity score	7.09 (9.76)	15.25 (10.42)	15.48 (9.7)	12.07 (9.9)	16.52 (10.4)
Symptom severity	%	%	%	%	%
PDS-score 1–10 Mild symptomatology	76.1	38.9	30.2	42.9	33.4
PDS score 11–20 Moderate symptomatology	12.1	28.5	24.2	42.8	23.9
PDS score 21–35 Moderate to severe symptomatology	20.6	27.6	21.3	3.6	33.5
PDS score > 36 Severe symptomatology	0.9	5.3	18.2	7.2	4.8

* Measured by the Post-traumatic Stress Diagnostic Scale (PDS).

**Table 2 ijerph-19-01783-t002:** Inter- and intra-group differences in coping intentions as measured by the Brief-COPE Questionnaire.

Coping Intentions	Intergroup Differences					Intragroup Differences			
	Asylum-Seekers	Participants without Migration Background							
	*M (SD)*	*M (SD)*	*F*	*p*	Partial *η*^2^		*M (SD)*	*F*	*p*
Active coping	3.39 (0.72)	3.39 (0.54)	1.31	0.243	0.057	Eritrea	3.61 (0.46)	2.00	0.061
						Somalia	3.50 (0.56)		
						Cameroon	3.07 (0.90)		
Denial ^e,f^	2.56 (1.05)	1.51 (0.61)	9.26	<0.001	0.309	Eritrea	2.55 (0.91)	1.07	0.402
						Somalia	2.80 (1.13)		
						Cameroon	2.61(1.08)		
Substance use	1.33 (0.69)	1.36 (0.53)	1.17	0.323	0.051	Eritrea	1.32 (0.57)	0.51	0.859
						Somalia	1.18 (0.67)		
						Cameroon	1.11 (0.29)		
Use of instrumental support ^c^	3.08 (0.82)	3.24 (0.69)	1.77	0.086	0.076	Eritrea	3.14 (0.58)	0.59	0.797
						Somalia	3.28 (0.68)		
						Cameroon	3.07 (0.96)		
Use of emotional support ^c^	3.21 (0.70)	3.04 (0.69)	2.64	0.009	0.109	Eritrea	3.32 (0.66)	1.05	0.420
						Somalia	3.28 (0.72)		
						Cameroon	2.69 (0.84)		
Humor ^c^	1.63 (0.81)	1.74 (0.65)	1.25	0.273	0.055	Eritrea	1.91 (0.83)	0.99	0.461
						Somalia	1.33 (0.63)		
						Cameroon	1.61 (0.88)		
Acceptance ^a,b,d^	2.32 (1.04)	2.08 (0.74)	1.23	0.284	0.054	Eritrea	1.95 (0.82)	1.94	0.070
						Somalia	2.95 (1.02)		
						Cameroon	2.14 (1.10)		
Religion ^a,b^	3.28 (0.99)	1.68 (0.84)	21.35	<0.001	0.498	Eritrea	3.50 (0.82)	2.48	0.021
						Somalia	3.63 (0.79)		
						Cameroon	2.46 (0.99)		

^a^ significant influence of religion in asylum-seekers; ^b^ significant influence of religion in Germans without a migration background; ^c^ significant influence of gender in Germans without a migration background; ^d^ significant influence of education in Germans without a migration background; ^e^ significant influence of symptom load in asylum- seekers; ^f^ significant influence of symptom load in Germans without a migration background.

**Table 3 ijerph-19-01783-t003:** Correlation analysis of the coping intentions and sociodemographic variables in asylum-seekers ^a^.

	Number of Traumatic Events	Posttraumatic Symptom Severity Score (PTSD)	Years of Formal Education	Educational Attainment
Active coping	0.06	0.02	0.05	0.07
Denial	0.11	**−0.21 ***	−0.14	**−0.19 ***
Substance use	0.01	**0.30 ***	**−0.20 ***	−0.16
Use of instrumental support	**−0.21 ***	**−0.21 ***	0.08	0.05
Use of emotional support	0.09	−0.14	−0.04	0.01
Humor	−0.01	0.17	0.02	0.05
Acceptance	−0.04	0.00	−0.15	**−0.23 ***
Religion	−0.05	−0.07	−0.06	−0.04

^a^ * *p* < 0.05. Values in bold show significant correlations.

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
