# Peer review of "Coping with Trauma and Symptoms of Post-Traumatic Stress Disorder: Exploring Intentions and Lay Beliefs about Appropriate Strategies among Asylum-Seeking Migrants from Sub-Saharan Africa in Germany"

_ijerph, 2022, doi:10.3390/ijerph19031783_

Round 1

Reviewer 1 Report

Dear Authors,

I have no objections to the substantive layer of the submitted text. I appreciate the application of both quantitative and qualitative approaches. I have a few methodological doubts.

In quantitative part of article, the three subgroups are compared (Eritrea, n = 41, Somalia n=36, Cameroon, n=25, other countries were not included). Among the statistical test used for comparison there is no test insensitive to groups unequality. Both ANOVA and T-test needs the assumptions of groups equality. Do these beetwen group differences/or lack of it not result from the different size of the compared groups?

Please, elaborate a bit more precise  the approach used during coding and data interpretation process. The IPA strategy concerns the way how data was interpreted, but there is no mention how the data was constructed. Especially beetwen the transcription stage and interpreting process. How the coding procedure was conducted, i.e. types of coding, codebook use, any automation process were used? Please give more insight on it to make this stage of the work more transparent.

Best regards,

Your peer reviewer.

Author Response

I have no objections to the substantive layer of the submitted text. I appreciate the application of both quantitative and qualitative approaches. I have a few methodological doubts.

In quantitative part of article, the three subgroups are compared (Eritrea, n = 41, Somalia n=36, Cameroon, n=25, other countries were not included). Among the statistical test used for comparison there is no test insensitive to groups unequality. Both ANOVA and T-test needs the assumptions of groups equality. Do these beetwen group differences/or lack of it not result from the different size of the compared groups?

Response: We appreciate the review of our manuscript and are thankful for your suggestions. Moreover, we would like to thank you for your positive feedback.

Thank you for raising this important point! We added a paragraph in the limitations section (line 610ff) by notifying readers that we quantitatively explored intragroup differences within the sample of asylum seekers with unequal sample sizes of each group. While equal sample size is not one of the assumptions made in a MANCOVA, there is a potential of reduced statistical power. Therefore, caution is warranted while interpreting the results. However, we advise readers to be aware of the fact that these analyses are exploratory in nature only and should not be considered principal analyses.  

Please, elaborate a bit more precise  the approach used during coding and data interpretation process. The IPA strategy concerns the way how data was interpreted, but there is no mention how the data was constructed. Especially beetwen the transcription stage and interpreting process. How the coding procedure was conducted, i.e. types of coding, codebook use, any automation process were used? Please give more insight on it to make this stage of the work more transparent.

Response: Thank you for raising this important point. We clarified the qualitative analysis and data interpretation in the section 3.2 Qualitative analysis.

Reviewer 2 Report

The work is an important contribution to the understanding of PTSD among asylum seekers/refugees in recent years in Europe. A better detailed description of the chosen sample criteria is missing (convenience sample why and with what consequences).

Another issue is the choice of origin countries that the authors take into consideration. Somalia, Eritrea and Cameroon are not the best representative origin countries in the atual moment (why not South-Sudan or DRC?) Those choices need to be further detailed.s

This work recognises the need for more research on the topic. Anyway the article is good and I would recommend it for publication.

Author Response

The work is an important contribution to the understanding of PTSD among asylum seekers/refugees in recent years in Europe. A better detailed description of the chosen sample criteria is missing (convenience sample why and with what consequences). Another issue is the choice of origin countries that the authors take into consideration. Somalia, Eritrea and Cameroon are not the best representative origin countries in the atual moment (why not South-Sudan or DRC?) Those choices need to be further detailed.

Response: We appreciate the review of our manuscript and are thankful for your suggestions. Moreover, we would like to thank you for your positive feedback.

Data collection for the study took place in 2016. We decided to target asylum seeking migrants from Sub-Saharan African countries, as the numbers of African migrants seeking for asylum in Germany have remained persistently high since 2009, with Eritrean and Somalian citizens being among the ten nationalities with the highest number of entries at that time (Federal Office for Migration and Refugees, 2018). We added this information for readers in line 129ff. In line 137ff, we added a paragraph describing our recruitment strategy in more detail and the inclusion criteria are described in line 162ff.

Reviewer 3 Report

The manuscript by F. Grupp and co-authors reports the results from a quantitative assessment of intentions of coping with trauma and post-traumatic stress disorder in people from several African countries seeking asylum in Germany and a qualitative study from focal groups on the religious, social and xxx coping strategies the participants use. It finds the importance of religion and faith, as well as of the opportunities to study and work, for asylum seekers to cope with past and present hardship. The analysis is in my opinion well designed and the manuscript is very clear and well-written. I only have some suggestions to improve some of the sentences, clarity of the tables and suggest further discussion of the results and their relevance to design strategies to help people to heal and incorporate to society.

My suggestions are:

Line 50 Speaks of “transition from post-traumatic distress to disorder”, leading to understand that “distress” and “disorder” are different levels or alteration. However, these terms have not been previously defined in the text It would be clearer if these are defined previously.

L 66: The sentence “With regard to cognitive coping strategies, unaccompanied minors from Sudan pointed out the utility of cognitive coping strategies…” would be clearer in this way: “With regard to cognitive coping strategies, studies with unaccompanied minors from Sudan pointed out the utility of cognitive coping strategies…”

L 71-73: The sentence “Further reported coping strategies include the suppression of traumatic memories and using distractions to avoid difficult thoughts and feelings, such as praying, sleeping, reading, playing a game, and talking to friends.” Would be clearer as: “Further reported coping strategies include the suppression of traumatic memories and using distractions such as praying, sleeping, reading, playing a game, and talking to friends to avoid difficult thoughts and feelings.”

METHODS:

It is not clear to me if the group without migration background also had exposure to trauma and/or PTSD? Otherwise, how was coping assessed in this group? and how are the groups paired to be compared?. It would be useful to clarify this early in the methods.

L 206: Delete “it” between “suggests that” and “ANOVA”

RESULTS

Table 2: It is not clear what the numbers reported refer to. It has an asterisk (coping intentions) that is not defined in the legend. Formatting is needed to include the word Cameroon in single lines (the n is in a different line).

Table 3 has a superindex (a) after the title, that is not defined in the legend.

L 328 Re-order sentence “During the focus group discussions, participants (n = 26) frequently referred to them-selves as “Africans” and members of their community.” As “During the focus group discussions, participants (n = 26) frequently referred to themselves and members of their community as “Africans”.”

DISCUSSION

L 426: add a colon between “origin” and “using”

The discussion in general describes the findings and states similarities of these with previous studies on the subject. However, it should emphasize the novelties found by this study. Otherwise, it seems like it does not provide new insights into the matter of study. It should also provide a sounder interpretation and dissertation on the findings, their importance in the context of the study and how these findings can be helpful. The comparison between the findings in asylum seekers and non migration background groups should also be discussed in the context of each of the groups, since they don´t seem to have many comparable situations.

Also, since the study finds that opportunities of going to school or having a job are good coping strategies, the authors could make a deeper dissertation on the role that policy makers can play to help migrants to cope with PTSD and adapt to their new circumstances, incorporating themselves to the german society.

Author Response

The manuscript by F. Grupp and co-authors reports the results from a quantitative assessment of intentions of coping with trauma and post-traumatic stress disorder in people from several African countries seeking asylum in Germany and a qualitative study from focal groups on the religious, social and xxx coping strategies the participants use. It finds the importance of religion and faith, as well as of the opportunities to study and work, for asylum seekers to cope with past and present hardship. The analysis is in my opinion well designed and the manuscript is very clear and well-written. I only have some suggestions to improve some of the sentences, clarity of the tables and suggest further discussion of the results and their relevance to design strategies to help people to heal and incorporate to society.

Response: We appreciate the review of our manuscript and are thankful for your suggestions. Moreover, we would like to thank you for your positive feedback.

Line 50 Speaks of “transition from post-traumatic distress to disorder”, leading to understand that “distress” and “disorder” are different levels or alteration. However, these terms have not been previously defined in the text It would be clearer if these are defined previously.

Response: Thank you for this remark. We revised the sentence in order to make it easier to understand for readers.

L 66: The sentence “With regard to cognitive coping strategies, unaccompanied minors from Sudan pointed out the utility of cognitive coping strategies…” would be clearer in this way: “With regard to cognitive coping strategies, studies with unaccompanied minors from Sudan pointed out the utility of cognitive coping strategies…”

Response: Thank you for the comment. We have modified the sentence.

L 71-73: The sentence “Further reported coping strategies include the suppression of traumatic memories and using distractions to avoid difficult thoughts and feelings, such as praying, sleeping, reading, playing a game, and talking to friends.” Would be clearer as: “Further reported coping strategies include the suppression of traumatic memories and using distractions such as praying, sleeping, reading, playing a game, and talking to friends to avoid difficult thoughts and feelings.”

 Response: Thank you for the comment. We have modified the sentence.

METHODS:

It is not clear to me if the group without migration background also had exposure to trauma and/or PTSD? Otherwise, how was coping assessed in this group? and how are the groups paired to be compared?. It would be useful to clarify this early in the methods.

Response: Thank you for raising these important points. Apparently, we did not sufficiently explain the underlying methodology and theoretical framework in the manuscript. The present study is placed within an illness explanatory model framework using a vignette methodology, promoting the perspective of laypeople. In this regard, we aimed to explore culturally and socially shared perspectives regarding appropriate coping strategies for trauma and PTSD, by extending the explanatory model approach beyond just the patient’s perspective. Therefore, we applied a case vignette design, following a paradigm widely applied in cross-cultural research in the fields of explanatory models of health and illness representations. In each part of the study, participants were given a standardized vignette (see supplementary material) illustrating a hypothetical friend describing symptoms of PTSD according to criteria outlined in the (ICD-10). Afterwards they were given the questionnaires with the instruction “What would you do in your friend’s place if you had these complaints?”. We do now elaborate more on this topic in line 152ff, line 164ff, and 185f .

L 206: Delete “it” between “suggests that” and “ANOVA”

 Response: Thank you for the suggestion for correction.

RESULTS

Table 2: It is not clear what the numbers reported refer to. It has an asterisk (coping intentions) that is not defined in the legend. Formatting is needed to include the word Cameroon in single lines (the n is in a different line).

Response: Thank you for this remark. We took out the asterisk and added in the table heading that coping intentions were measured by the Brief COPE. Additionally, we have formatted the column with the countries of origin of the asylum seekers.

Table 3 has a superindex (a) after the title, that is not defined in the legend.

Response: The legend of table 3 has moved to the next page during formatting.

L 328 Re-order sentence “During the focus group discussions, participants (n = 26) frequently referred to them-selves as “Africans” and members of their community.” As “During the focus group discussions, participants (n = 26) frequently referred to themselves and members of their community as “Africans”.”

Response: We have modified the sentence.

DISCUSSION

L 426: add a colon between “origin” and “using”

 Response: Thank you for the suggestion for correction.

The discussion in general describes the findings and states similarities of these with previous studies on the subject. However, it should emphasize the novelties found by this study. Otherwise, it seems like it does not provide new insights into the matter of study. It should also provide a sounder interpretation and dissertation on the findings, their importance in the context of the study and how these findings can be helpful. The comparison between the findings in asylum seekers and non migration background groups should also be discussed in the context of each of the groups, since they don´t seem to have many comparable situations.

Also, since the study finds that opportunities of going to school or having a job are good coping strategies, the authors could make a deeper dissertation on the role that policy makers can play to help migrants to cope with PTSD and adapt to their new circumstances, incorporating themselves to the german society.

Response: Thank you for your comment and the important advice. We have revised the discussion again and integrated the additional information.
